# Pre-Hydrated Berry Pomace in Wheat Bread: An Approach Considering Requisite Water in Fiber Enrichment

**DOI:** 10.3390/foods9111600

**Published:** 2020-11-03

**Authors:** Anne-Marie Reißner, Amanda Beer, Susanne Struck, Harald Rohm

**Affiliations:** Chair of Food Engineering, Technische Universität Dresden, 01062 Dresden, Germany; amanda.beer@web.de (A.B.); susanne.struck@tu-dresden.de (S.S.); harald.rohm@tu-dresden.de (H.R.)

**Keywords:** pomace, dough, bread, water absorption, rheology, fiber, texture, sustainability, proofing

## Abstract

Using blackcurrant pomace, for instance, in bread, prevents wasting this by-product of fruit juice production, thereby also taking advantage of its nutritional benefits. This study investigated the effects of blackcurrant pomace incorporation in wheat dough and the quality of the resulting breads. Two concepts were addressed: (a) adjusting the water level based on the dough consistency and (b) preparing pre-hydrated pomace by applying the determined water content prior to dough preparation and using it for bread making. Samples with wholegrain spelt flour were used for additional comparison. This study revealed that instant pomace incorporation with an adjusted water level diminished the dough stickiness and baking loss, but resulted in stiffer dough with delayed proofing and a decreased bread volume. The kneading resistance pointed to continued swelling after kneading, concomitant with a lower amount of available free water. Counteracting the competition for water of the flour components and pomace fiber by applying pre-hydrated pomace turned out to be successful. The deteriorating effects were reduced to a larger extent by pomace hydrated in hot water. Despite a similar composition, the products of wholegrain spelt flour deviated from the pomace formulations as well as from wheat breads (producing the highest water absorption but smallest loaves). As the water absorption of pomace fiber largely influences the product properties, a pre-hydration of pomace to be included in wheat dough can be highly recommended to enhance processing properties and product quality.

## 1. Introduction

The by-product of fruit juice processing contains valuable nutritional compounds that are lost when the pomace is disposed of. In addition to the dominating dietary fiber fraction mainly comprising pectin, hemicellulose, cellulose, and lignin, blackcurrant pomace contains considerable amounts of protein, fat, and polyphenols [1,2]. In line with the desire for a healthy and sustainable diet, incorporating this by-product in foods contributes to the enhancement of their nutritional and sensory properties [3]. However, the application in bakery products triggers a number of technological challenges.

In their review, Martins et al. [4] summarized that supplementing wheat bread with dietary fiber affects dough development time and stability, increases dough consistency, and therefore causes difficulties in processing. Bread rich in dietary fiber usually shows a reduced volume during baking and exhibits a firm, less cohesive crumb. Interactions between fiber and wheat protein as well as gluten dilution have been mentioned as the main reasons for these effects [4,5]. As regards soluble fiber, hydrocolloids such as hydroxypropylmethylcellulose, sodium alginate, or xanthan increased bread volume and crumb moisture at application levels even below 1% [6], inulin improved gas retention up to 3% [7], and pectin and β-glucan at levels of 1% enhanced crumb structure without affecting sensory acceptability [8]. By contrast, as little as 2% of insoluble cellulose impaired dough extensibility and thus gas retention capacity and bread volume [9].

The high water-binding capacity of dietary fiber can be attributed to the large number of hydroxyl groups. In wheat dough, they compete with flour for water. This affinity potentially limits starch swelling and hence gelatinization [10]. In several studies, fruit pomace or isolated dietary fiber was incorporated in wheat dough without taking its high water-binding capacity into account. Only a few authors discussed the hydrating properties of the fiber and considered them during bread making [7,11,12]. Aiming at a similar dough consistency by adapting kneading resistance in farinograms [5] or by adjusting the storage modulus in rheological experiments through targeted water addition [13] turned out to be a successful way to improve the resulting product properties. However, to date, only short time scales after water addition, i.e., during kneading or after a resting period of 10 min [5,13], have been considered in analysis, while determining the water-binding capacity takes 30 min or longer.

In practice, bakeries usually apply seeds, grains, and groats in a pre-hydrated state. This swelling is performed in cold or hot water, and generally enhances water binding during processing. When hot water was used, enzyme activities were reduced in pre-hydrated grains [14].

Previously, we studied the interactions of berry pomace and wheat macromolecules in model dough systems, containing solely flour, pomace, and water [5]. It was hypothesized that pomace levels below 20% would produce bread with satisfying characteristics, but farinograms pointed to a prolonged dough development time and increased water absorption [5]. The aim of the current study was to systematically analyze the dough processibility and the properties of bread enriched with blackcurrant pomace that was subjected to pre-hydration at different conditions. Since mixtures of wheat flour with 10% blackcurrant pomace have similar contents of fiber, fat, and protein to wholegrain spelt flour, this common resource for fiber enrichment was included for additional comparison. The pomace used is a representative example of a fiber-rich, zero-waste by-product, and general indications for enrichment with other fibers may be derived from the results.

## 2. Materials and Methods

### 2.1. Materials

The ingredients used for bread making were demineralized water, all-purpose wheat flour type 550 (Dresdener Mühle, ZN der PMG Premium Mühlen Gruppe GmbH & Co., Dresden, Germany), fresh yeast (Uniferm GmbH & Co. KG, Werne, Germany), sodium chloride (Carl Roth GmbH & Co. KG, Karlsruhe, Germany), and wholegrain spelt flour (Frießinger Mühle GmbH, Bad Wimpfen, Germany). Blackcurrant pomace (Döhler GmbH, Neuenkirchen Hadeln, Germany) was processed to a powder (volume-based median of particle size x_50,3_ = 187.4 ± 9.2 µm) by simultaneous drying and milling (Mahltechnik Görgens GmbH, Dormagen, Germany) at a product temperature <50 °C. The water-binding capacity of the pomace powder was analyzed by adding a specified amount of water, hydrating for 30 min, and decanting excess water after centrifugation [2], and is expressed as the amount of water bound per mass unit of dry pomace powder. The moisture content was determined by drying to a constant mass at 103 °C for pomace powder and at 130 °C for wheat flour according to ICC standard 110/1 [15]. The composition of the flours as specified by the suppliers, and of the blackcurrant pomace as previously published by Reißner et al. [2], is given in Table 1.

### 2.2. Bread-Making Procedure

The reference dough (Ref) consisted of 3 g of yeast and 1.5 g of NaCl dissolved in 56.5 g of deionized water per 100 g of wheat flour (corrected to 140 g/kg of moisture). The wheat flour was partially replaced by 10% pomace powder (10P) or, for additional comparison and because wholegrain spelt (WS) flour shows a similar composition to 10P (Table 1), completely substituted with WS flour. To ensure controlled dough handling, the water content in each system was adjusted in preliminary experiments according to ICC standard 115/1 [16] with slight modifications, aiming at achieving a defined dough consistency of 600 Brabender Units (BU). Two modifications of 10P were prepared by pre-hydrating the pomace powder for 30 min in water at 20 °C (10Pc) or mixing with boiling water (10Ph), using the same amount of water as was determined by water absorption for 10P (59.1 g/100 g of flour). The 10Ph water reached room temperature during the soaking period, and the amount of water that evaporated was added before kneading.

Dough was prepared in a 300 g farinograph cell at 26 °C (Brabender GmbH & Co. KG, Duisburg, Germany), first blending the flour (Ref) or the dry flour/pomace mixture (10P) for 1 min at 63 rpm. After adding the remaining ingredients (including the pre-hydrated pomace for 10Pc or 10Ph), the dough was kneaded for 4 min. During pre-proofing at 26 °C, the dough was kneaded after 15 and 30 min for 15 s at 63 rpm, subsequently shaped into pieces of 100 g, and evenly perforated with a pin roller. After 35 min of main proofing in a proofing chamber (32 °C, 80% relative humidity), the bread was baked for 20 min at 240 °C (MIWE Condo, Michael Wenz GmbH, Arnstein, Germany). The loaves were allowed to cool for 30 min at 20 °C in a Peltier-cooled IPP-55 incubator (Memmert GmbH & Co. KG, Schwabach, Germany), packed in polyethylene bags, and analyzed within 3 h. Two independent batches of 4 loaves were produced for each formulation.

### 2.3. Dough Analysis

Expansion during the main proofing of 30 g of shaped dough (formulations: Ref, 10P, 10Pc, and 10Ph) was monitored in graduated cylinders over a period of 60 min at 32 °C and 80% humidity (*n* = 2). Changes in volume were plotted against the proofing time and fitted by cubic equations.

The pH was measured immediately after dough preparation by means of a pH meter (Knick Elektronische Messgeräte GmbH & Co. KG, Berlin, Germany) with a puncture electrode for viscoelastic samples (BlueLine, Xyleme Analytics Sales GmbH & Co. KG, Germany), in triplicate (*n* = 6).

The dough resistance and extensibility were determined using a TA.XTplus Texture Analyzer (Stable Micro Systems Ltd., Godalming, UK) and a Kieffer dough extensibility rig by following the procedure described by Verheyen et al. [17]. Immediately after preparation, 20 g of dough was round shaped, formed into strands with the aid of an oiled Teflon sample plate, and allowed to rest for 10 min at 32 °C. Five strands of each dough (*n* = 10 replicates per formulation) were extended with a rig velocity of 3.3 mm/s until tearing. The maximum force refers to the dough resistance, and the travelling distance of the rig until tearing is defined as the extensibility. To account for time-dependent effects, additional measurements were performed after pre-proofing (30 min) and main proofing (65 min of total proofing time). The respective dough samples were kneaded twice as in the bread-making procedure (i.e., after 15 and 30 min).

Dough stickiness was defined as the maximum tensile force when removing a cylindrical probe from a defined dough surface. The measurements were performed using a Chen-Hoseney cell mounted on the TA.XTplus Texture Analyzer as described by Struck et al. [5]. Ten measurements per dough (*n* = 20) were recorded after 75 min of main proofing, as the dough stickiness, when hydration processes are considered as being complete, is usually independent of the fermentation time [18].

For small strain oscillation rheology, dough with an identical formulation was prepared separately without yeast. Dough with wholegrain spelt was, due to its stickiness, excluded from the rheological experiments. A dough sample of 8.5 g was rolled out and placed on the lower plate of a Physica MCR 300 rheometer (Anton Paar GmbH, Graz, Austria). A 50 mm-diameter serrated upper plate was lowered until a gap of 2 mm was reached. Prior to measurements, excess material was trimmed, the gap was covered with Vaseline oil, and the sample was allowed to relax in the gap for 5 min at 25 °C. The measurement regime comprised of a frequency sweep from 10 to 0.1 Hz in the linear viscoelastic region at a strain of 0.3% (10 points per decade) at 25 °C, followed by a temperature sweep up to 100 °C at 6.3 K/min, representing the heating rate during oven baking, and a frequency of 1 Hz. The storage modulus (G′) and loss modulus (G″) were derived from the measurements. The complex modulus G* is the Pythagorean sum of G′ and G″ and refers to the overall resistance to deformation and, hence, stiffness of the dough. The loss tangent (tan δ = G″/G′) indicates whether elastic or viscous contributions are dominating. All the measurements were performed on two individually prepared doughs (*n* = 2).

### 2.4. Bread Analysis

The relative baking loss was calculated from mass difference of 8 loaves before and after baking. The bread volume was determined using the canola seed method 10-05.01 [19]. The pH of the crumb was measured with a puncture electrode (two breads in duplicate) as described for dough.

The crust firmness was determined with the TA.XTplus Texture Analyzer by penetrating the surfaces of 4 loaves at 8 individual points with a cylinder of 4 mm diameter at a testing speed of 40 mm/s. The maximum force was defined as the crust firmness (*n* = 32). The breads were then cut into 12.5 mm-thick slices, and the crumb firmness of two stacked central slices was measured by compression with a cylinder of 25 mm diameter. In line with approved method 74-09.01 [20], the testing speed was 1 mm/s and the final compression was 40%. The compression force at 6.25 mm (25% compression) was defined as the crumb firmness. Two slices of four breads were analyzed in quadruplicate (*n* = 32).

For analyzing the crumb structure, eight bread slices were scanned with a CanoScan Lide 110 (Canon Germany GmbH, Krefeld, Germany) at a resolution of 600 dpi. Using the ImageJ software (National Institutes Health, Bethesda, MD, USA), the images were converted to 8-bit grayscale, cropped to 30 × 30 mm^2^, and binarized by applying the Otsu threshold algorithm. Cell counts and respective cell areas (mm^2^) were determined, and the cell density (cells/cm^2^) and total cell area (mm^2^/100 mm^2^) were calculated.

For the following analyses, 100 g of crumb (representative sample of 2 breads per batch) was comminuted for 10 s at 5000 rpm in a Grindomix GM200 mill (Retsch GmbH, Haan, Germany). The crumb moisture (*n* = 6) was determined at 105 °C in a MA30 moisture analyzer (Sartorius AG, Göttingen, Germany). The color was measured with a Luci 100 spectral colorimeter (D65 xenon lamp, 10° observer; Hach Lange GmbH, Düsseldorf, Germany) after compressing 2 g of powdered crumb with a 32 mm acryl plunger in a Quartz glass cylinder (d = 34 mm). The color primaries of quadruplicate measurements were transferred into the CIE-Lab color space, and the lightness L*, hue angle h_ab_ (describing the color quality), and chroma C* (indicating saturation) were calculated for interpretation [21].

### 2.5. Statistical Analysis

Analysis of variance (ANOVA) with subsequent Student–Newman–Keuls and Duncan post hoc testing at *p* ≤ 0.05 (*p* ≤ 0.1 for crumb firmness), and regression analyses were performed with SAS University Edition 6p.2 (SAS Institute Inc., Cary, NC, USA). Before analysis, biological and technical replicates were pooled.

## 3. Results and Discussion

### 3.1. Hydration Properties

The amount of water necessary to achieve 600 BU in the reference dough was 56.5 mL per 100 g of wheat flour. This amount increased by 4.6% to 59.1 mL/100 g for 10P (10% of the wheat flour replaced by blackcurrant pomace), reflecting its higher water affinity. The dough prepared from wholegrain spelt flour absorbed even more water (67.1 mL/100 g). Frakolaki et al. [22] attributed this water affinity in farinograms to a higher protein content compared to wheat flour, which is also true for berry pomace. The high dietary fiber content in wholegrain flour is another factor that influences water absorption.

After 4 min of kneading, the resistance of the Ref dough was 621 ± 14 BU. As reported previously [5], the incorporation of dry pomace prolonged dough development but resulted in a similar dough resistance after an identical kneading time (605 ± 5 BU). On the other hand, dough containing pre-hydrated pomace showed a significantly higher resistance of 712 ± 12 and 691 ± 30 BU for 10Pc and 10Ph, respectively. This phenomenon emphasizes the time-dependent process of hydration: dietary fiber and starch granules soaked up water beyond the 4 min kneading time; hence, prolonged hydration periods seem to be reasonable. This means that pre-hydration could be beneficial for dough formation, whereas extending kneading time probably results in a damaged gluten structure. The WS dough showed an intermediate kneading resistance (645 ± 1 BU), less pronounced than that for pre-hydrated pomace dough, and, concomitantly, prolonged dough development.

The dry matter-related water-binding capacity of the blackcurrant pomace powder was 4.46 g/g at room temperature (20 °C) and increased to 5.70 g/g after hydrating the pomace in boiling water. This can be attributed to the denaturation of pomace proteins and to the gelation of water-soluble pectins after cooling, which still make up 10% in enzymatically treated blackcurrant pomace [1]. In light of the increased kneading resistance and water binding capacity for 10Ph, even more water for dough preparation may be feasible, but that would require a different methodology. It should further be noted that such an adaption may directly affect dough and bread properties.

### 3.2. Rheology during Simulated Baking

Small-amplitude oscillation rheology provided further insights into the effects of pomace incorporation. Figure 1 depicts the complex modulus (G*) and the loss tangent (tan δ) as observed in frequency sweeps at 25 °C, and in subsequent temperature sweeps. All the doughs showed predominantly elastic behavior (tan δ < 1).

At 25 °C, the dough stiffness increased with increasing frequency, with a power law slope of ≈0.2, thus pointing to shear-thinning behavior. Compared to Ref, the addition of 10% pomace resulted in much stiffer doughs. This stiffening indicates that cross-linking between pomace compounds and wheat proteins occurs, and that less free water is available in the dough [23]. As compared to 10P, 10Pc did not show a significant impact on dough rheology, but 10Ph shifted G* towards Ref. The between-dough differences in tan δ were mainly caused by differences in the storage modulus, as the loss moduli were almost similar. It can be assumed that the interactions of dietary fiber with wheat proteins remained unchanged, but denatured pomace proteins released bound water and increased molecular mobility. Alba et al. [10] analyzed the rheology of wheat dough with blackcurrant pomace and its insoluble dietary fiber fractions (5–20%) without adjusting the water level in formulations, and attributed the observed stiffening to higher cellulose contents. This effect became more prominent at higher pomace fractions.

During baking simulated in temperature sweeps, three stages can be distinguished. Through heating, G* reached a minimum in the first stage, and was the lowest for Ref (5 kPa) and approx. three times higher for 10P (14 kPa). The differences in the temperature at this minimum (52.6–55.2 °C) were not significant. The reduced stiffness can be explained by thermally activated molecules, a softened protein structure, and melting of the fat [13]. Further heating induced starch swelling, gelatinization, and protein denaturation [13], leading to reduced molecular mobility. The tan δ started to drop at 73 °C, representing the onset of major structural changes, evident from increased elastic contributions. The maximum stiffnesses were 94 and 144 kPa for Ref and 10P, respectively. The 10Ph (112 kPa) dough was similar to Ref, indicating continuing effects of the previous swelling on the dough stiffness. The corresponding temperature at the maximum G* tended to decrease in the order Ref, 10P, 10Pc, and 10Ph (87.3–84.0 °C), pointing to reduced water competition with starch granules when hydrated pomace was incorporated. In contrast to the findings of Alba et al. [10], the gelation temperature was not affected by pomace application. The reduced starch content of the pomace-containing dough was reflected by a less pronounced magnitude of changes in stiffness between the minimum G* and maximum G* [24].

### 3.3. Changes in Dough Properties during Proofing

The increase in dough volume during the main proofing, starting at 30 min after kneading, is depicted in Figure 2. The curves were fitted by cubic equations, achieving reliable coefficients of determination R^2^ > 0.99.

After an initial phase of approx. 10 min, the volume increased steadily. The quadratic coefficient indicates the highest rise for Ref and 10Ph, but the slope declined in the final stage of proofing. To identify effects on bread volume, the Ref dough was baked after different proofing times. The highest bread volume was observed after 35 min of main proofing, and this time was therefore chosen as the proofing time in all other baking experiments (i.e., a total fermentation time of 65 min). Comparing with the increase in dough volume, this optimized bread volume was reached after the inflection point of the respective cubic function (27 min) but before the dough volume curve flattened out. Sample 10P showed a delayed and impaired increase in volume, and the inflection point was shifted to 31 min. Pre-hydrated pomace in the dough attenuated this deteriorated proofing behavior, 10Ph to a larger extent than 10Pc.

Figure 3 shows the dough resistance and extensibility as affected by the proofing time. The resistance increased within 30 min after kneading, pointing to continuing structure formation. During the main rise, up to 65 min of total proofing time, enzyme activity is responsible for partial gluten degradation and stress relaxation [17], so the dough resistance significantly decreased. As opposed to Ref, 10P resulted in doughs with higher resistance after proofing for 30 and 65 min, presumably because of interactions between dietary fiber, wheat or pomace proteins, and polysaccharides [25,26]. This is also consistent with the increased stiffness in the frequency sweeps at 25 °C (Figure 1). Both pre-hydrated samples differed significantly from 10P. Whereas 10Pc showed a similar resistance to Ref, 10Ph exhibited a much higher resistance after pre-proofing. The dough resistance finally dropped after 65 min to 0.42 ± 0.05 N, which is lower than that of 10P. This behavior is caused by (a) the effects of protein denaturation caused by pre-hydrating pomace in boiling water, and (b) the available water that favored gluten relaxation. The dough resistance in WS increased continuously up to 65 min of proofing time. The higher gliadin levels in the wholegrain spelt flour retarded a tension drop [22], and firmer doughs were obtained after fermentation. However, WS showed the lowest dough resistance among all formulations.

The extensibility was the highest for all the doughs immediately after kneading and decreased with proofing time (Figure 3). The reference showed the highest extensibility, followed by WS, indicating a strong gluten network [27]. Pomace application generally decreased the extensibility, which can be attributed to fiber enrichment; a restricted molecular mobility through the formation of fiber/protein interactions, such as hydrogen bonds; and the dilution of the gluten network [5,27]. Nevertheless, immediately after kneading, the extensibility of 10Pc was comparable to that of Ref. Compared to dough resistance, it was observed that the higher extensibility in the pomace-containing dough was accompanied by a decreased resistance. Therefore, the low resistance in extension is presumably attributable to a weakened gluten structure that could affect bread volume [28].

The dough stickiness, reflecting free surface water, is displayed in Table 2. The highest stickiness was observed for Ref and WS. When handling WS, its stickiness appeared to be much higher than any other dough, which was not reflected by the measurements but can be attributed to the gliadin proteins [22]. Although the pomace dough contained more water than Ref, its water-binding capacity compensated the stickiness, which decreased significantly.

The impact of pomace pre-hydration turned out to be negligible. The acidity of the blackcurrant pomace, expressed as tartaric acid equivalent, was reported to be 4.5 g/kg [2]. Therefore, the dough pH dropped when pomace was added, namely, from 5.77 (Ref) to 5.11 (10P). As the reasonable pH for yeast fermentation ranges from 4.5 to 6.0 [29], negative effects on bread quality should not be expected.

### 3.4. Effects on Bread Characteristics

Chemical processes and water evaporation during baking reduced the acidity of the Ref bread crumb to pH 5.9. The typical acidity of wheat bread varies in the range of pH 5.2–6.0 [30]. By contrast, the acidity of the pomace dough increased through baking to approx. pH 5.0, due to altered water binding, which increased organic acids in the free water phase.

In terms of staling, using pomace could be beneficial, since losing moisture during storage could be counteracted through the elevated water-binding capacity of the fiber [31,32]. This presumption is supported by the reduced baking loss and enhanced crumb moisture (Table 2) in the pomace-containing bread. Pre-hydrated pomace affected water retention during the baking process, and 10Ph increased the crumb moisture to an even greater extent than 10Pc. It can be assumed that the hydrated fiber and pomace protein compete less for water with the flour components when previously heated. Furthermore, the reduced gas retention ability of 10Pc may have caused an additional moisture loss. WS lost more water during baking, since the crumb moisture was identical to 10Ph’s despite the highest water absorption of the wholegrain spelt flour during kneading.

When containing blackcurrant pomace, the bread volume decreased from 181.8 mL (Ref) to 162.2 mL (10P). WS resulted in even smaller loaves, as the baking loss was high and the baking properties of its gluten composition were different [22]. As opposed to 10P, 10Ph substantially increased bread volume, while 10Pc had negligible effects. Various studies have reported detrimental effects on bread quality from applying pomace or isolated fiber fractions [10,33,34]. The causes were attributed to diluted gluten and protein/fiber interactions [5,34]. However, appropriate amounts of soluble fiber [33] and, as shown in this study, adjusting the water content based on dough stiffness are helpful for minimizing undesired effects.

The crust and crumb firmness as influenced by wheat flour replacement is shown in Table 3. The main differences were apparent for WS, which solidified the bread structure. The 10P produced marginal changes compared to Ref; only the crumb firmness of 10Pc tended to be reduced compared to 10Ph. Other studies reported softer crumbs when applying up to 30% grape pomace [12], 20% wheat bran [11], or 3% pea or carob fiber [7] while taking the water absorption of the fiber into account.

As regards the bread structure parameters, supporting the texture data, Ref showed the lowest cell density and total cell area (26 cells/cm^2^; 26 mm^2^/100 mm^2^) (Table 3). For WS, the proportion of cells on the total area was increased (33 mm^2^/100 mm^2^), indicating a less compact structure with extended pores. A weakened gluten structure with a low gas holding capacity manifests in the aggregation of small cells into a few larger cells [14,35]. Alba et al. [10] reported a reduced cell number and cell area accompanied with a denser crumb for bread with 10% blackcurrant pomace. In this study, the cell density and total area of the pomace breads increased concurrently, contradicting crumb defects. The softened crumb of 10Pc showed the highest total cell area (38 mm^2^/100 mm^2^).

According to Holmes and Hoseney [36], the dough pH affects the gas production rate of the yeasts and correlates with the final bread volume and crumb grain. In this study, both the dough and bread pH correlated negatively with crumb cell density (*r* ≈ −0.97) but were not linked to the volume. In detail, a lower pH tended to slow down yeast activity, and smaller cells were formed. This resulted in a higher cell density. As the acidity of the dough was still appropriate, a high number of cells were formed during fermentation.

Comparing the dough rheology and bread texture, the differences in stiffness between the Ref wheat dough and pomace-containing dough were reduced after simulated baking, as well as the differences in bread firmness being less pronounced than the differences in dough resistance and extensibility. In total, analyzing the dough with rheology, combining frequency and temperature sweeps, can be considered as useful for predicting product handling and bread characteristics.

Regarding appearance and color (Figure 4), the lightness of the crumb of the pomace breads decreased from 65.0 to approx. 40.0, the color saturation/chroma from 13.3 to 2.9, and the hue angle from 86.1 to 52.5 (each from Ref to 10P). The dark purple pomace color changed during baking and resulted in grayish bread crumbs. The color of the pre-hydrated samples (10Pc and 10Ph) differed to that of 10P below the visually just-noticeable threshold (with ΔE* = 0.07 and 0.79, respectively). WS was characterized by L* = 54.7, C* = 15.1, and h_ab_ = 68.8. Just as the color of WS varied, the flour proteins and dietary fiber of the wholegrain spelt flour acted completely differently than the mixtures of wheat flour and 10% pomace. At present, consumers are getting accustomed, one by one, to dark-colored crumbs, such as black burger buns with natural squid ink. The appearance of blackcurrant breads is extraordinary but points to natural ingredients. Alba et al. (2020) investigated the aroma profile of blackcurrant pomace bread and detected >100 volatiles [10]. They attributed the terpene hydrocarbons to the berries and carbonyl compounds to lipids from their seeds [10]. Starting from this, a human sensory study would be fascinating.

## 4. Conclusions

In a previous study on dough microstructure, it was predicted that using blackcurrant pomace up to 10% could be successful [5]. In terms of the findings obtained in the current study, this assumption can be confirmed, especially when its high water-binding capacity is considered. The hydration properties of blackcurrant pomace affected the bread-making procedure substantially. Aiming at a similar dough consistency proved to be successful. Since pomace pre-hydration increased the kneading resistance in farinograms, even more addition of water seems to be feasible. A pre-hydration in hot water improved the dough handling properties and quality parameters of the resulting bread to a larger extent than pomace soaked at room temperature. Under the conditions examined, the pre-hydration of blackcurrant pomace with boiling water for 30 min is favorable. The appropriate water amount can be determined by aiming at a similar kneading resistance in farinograms.

The main outcome of the study is that considering the water-binding capacity of fruit fiber is of particular relevance in bread-processing technology but also for research. A similar water content, but different hydration strategies, provoked varying dough and bread properties. As to date, the hydration of dietary fiber has often been neglected or even ignored, this is an important finding.

The presented approach may help to incorporate higher fractions of dietary fiber in bakery products. The incorporation of blackcurrant pomace is only one promising example. The enzyme activity of pre-hydrated pomace is an interesting aspect to be focused on in prospective studies.

## Figures and Tables

**Figure 1 foods-09-01600-f001:**
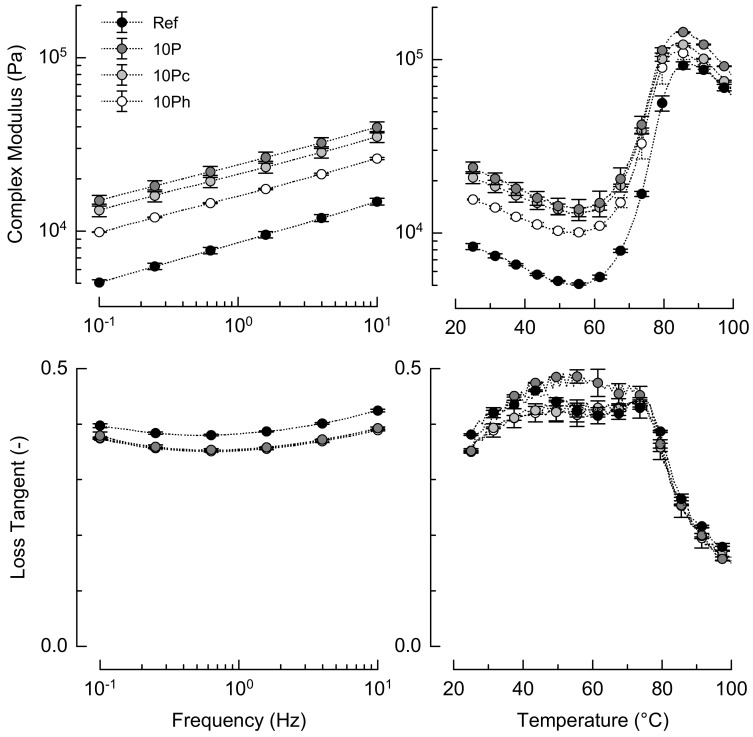
Frequency dependency of dough stiffness and the loss tangent at 25 °C (left), and development of stiffness and loss tangent during simulated baking (right) (*n* = 2). Ref, reference dough; 10P, dough based on 90% wheat flour and 10% pomace; 10Pc, similar to 10P but with pomace soaked in cold water; 10Ph, similar to 10P but with pomace soaked in boiling water. To improve clarity, not all data points are plotted.

**Figure 2 foods-09-01600-f002:**
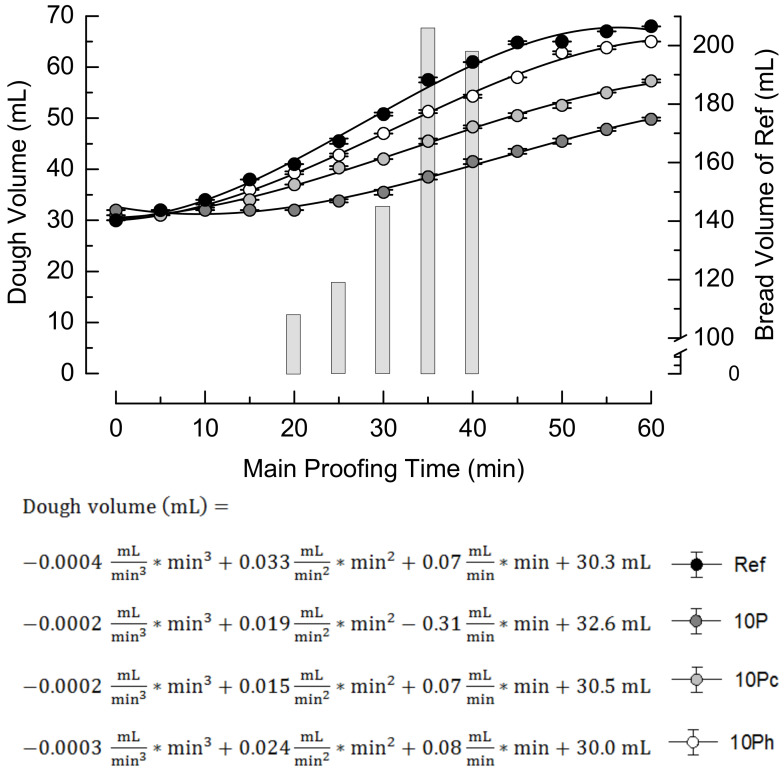
Increase in dough volume during main proofing (symbols and curves, *n* = 2) and volume of the reference bread baked after different main proofing periods (bars, *n* = 4). Ref, reference dough; 10P, dough based on 90% wheat flour and 10% pomace; 10Pc, similar to 10P but with pomace soaked in cold water; 10Ph, similar to 10P but with pomace soaked in boiling water.

**Figure 3 foods-09-01600-f003:**
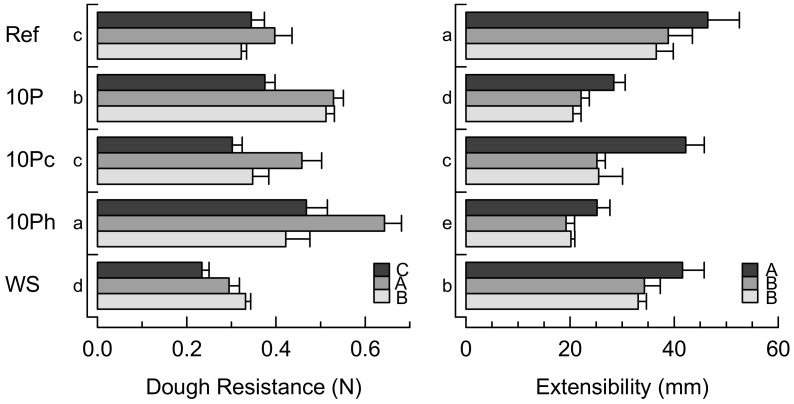
Dough resistance (left) and extensibility (right) of dough after different periods of proofing: dark gray, immediately after kneading; gray, after 30 min pre-proofing; light grey, after 30 min pre-proofing + 35 min main proofing. Ref, reference dough; 10P, dough based on 90% wheat flour and 10% pomace; 10Pc, similar to 10P but with pomace soaked in cold water; 10Ph, similar to 10P but with pomace soaked in boiling water; WS, wholegrain spelt flour dough. Lowercase letters indicate significant differences (*p* < 0.05) among formulations; uppercase letters indicate significant differences (*p* < 0.05) among proofing times.

**Figure 4 foods-09-01600-f004:**
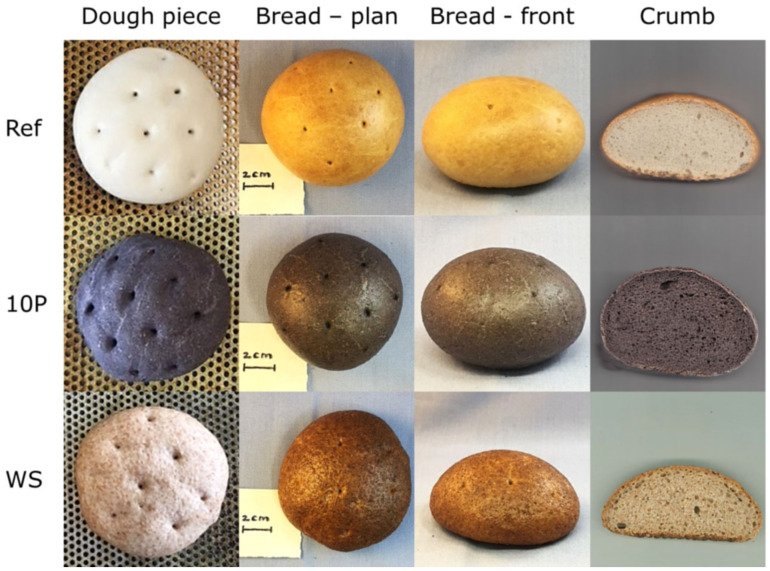
Appearance of dough, bread, and crumb. Ref, reference; 10P, dough/bread based on 90% wheat flour and 10% pomace; WS, wholegrain spelt flour.

**Table 1 foods-09-01600-t001:** Dry matter-based proximate composition of raw materials.

Component (g/100 g)	Blackcurrant Pomace ^1^	Wheat Flour	90% Wheat Flour + 10% Pomace	Wholegrain Spelt Flour
Carbohydrates	2.2	81.2	73.3	70.4
Dietary fiber	59.1	2.7	8.3	9.3
Protein	15.7	15.1	15.1	15.7
Fat	20.2	0.9	2.9	2.1
Ash	2.7	0.7	0.9	0.9

^1^ Data from Reißner et al. [2].

**Table 2 foods-09-01600-t002:** Dough and bread characteristics.

Sample ^1^	Dough Stickiness (N) (*n* = 20) ^2^	Dough pH (-) (*n* = 6) ^2^	Bread pH (-) (*n* = 4) ^2^	Baking Loss (%) (*n* = 8) ^2^	Bread Volume (mL) (*n* = 8) ^2^	Crumb Moisture (g/100 g) (*n* = 6) ^2^
Ref	0.49 ± 0.03 ^a^	5.77 ± 0.01 ^a^	5.90 ± 0.01 ^a^	15.59 ± 0.38 ^b^	181.83 ± 4.50 ^a^	41.41 ± 0.15 ^d^
10P	0.35 ± 0.05 ^b^	5.11 ± 0.01 ^d^	5.01 ± 0.01 ^c^	14.34 ± 0.20 ^cd^	162.19 ± 5.89 ^c^	41.68 ± 0.18 ^c^
10Pc	0.35 ± 0.04 ^b^	5.15 ± 0.01 ^c^	5.01 ± 0.01 ^c^	14.61 ± 0.36 ^c^	163.45 ± 2.33 ^c^	42.09 ± 0.19 ^b^
10Ph	0.32 ± 0.05 ^b^	5.15 ± 0.01 ^c^	5.01 ± 0.01 ^c^	14.10 ± 0.28 ^d^	171.71 ± 7.89 ^b^	42.78 ± 0.11 ^a^
WS	0.50 ± 0.05 ^a^	5.74 ± 0.01 ^b^	5.72 ± 0.01 ^b^	17.45 ± 0.24 ^a^	141.25 ± 0.45 ^d^	42.85 ± 0.09 ^a^

^1^ Ref, reference dough/bread; 10P, dough/bread based on 90% wheat flour and 10% pomace; 10Pc, similar to 10P but with pomace soaked in cold water; 10Ph, similar to 10P but with pomace soaked in boiling water; WS, dough/bread of wholegrain spelt flour. ^2^ Mean values (± standard deviation) in a column with different superscripts differ significantly (*p* < 0.05).

**Table 3 foods-09-01600-t003:** Bread texture as affected by wheat flour replacement.

Sample ^1^	Crust Firmness (N) (*n* = 32) ^2^	Crumb Firmness (N) (*n* = 16) ^3^	Crumb Cell Density (cells/cm^2^) (*n* = 4) ^2^	Total Cell Area (mm^2^/100 mm^2^) (*n* = 4) ^2^
Ref	20.87 ± 1.40 ^b^	5.49 ± 1.46 ^bc^	26.31 ± 4.45 ^b^	25.80 ± 3.15 ^c^
10P	22.09 ± 2.81 ^ab^	5.01 ± 0.59 ^bc^	35.97 ± 4.31 ^a^	33.53 ± 2.29 ^b^
10Pc	22.20 ± 3.33 ^ab^	4.81 ± 0.79 ^c^	32.75 ± 5.53 ^ab^	38.18 ± 2.70 ^a^
10Ph	22.57 ± 3.53 ^ab^	6.02 ± 0.88 ^b^	34.92 ± 5.31 ^a^	36.45 ± 2.65 ^ab^
WS	23.51 ± 3.30 ^a^	13.71 ± 3.32 ^a^	26.06 ± 0.39 ^b^	32.98 ± 2.24 ^b^

^1^ Ref, reference dough/bread; 10P, dough/bread based on 90% wheat flour and 10% pomace; 10Pc, similar to 10P but with pomace soaked in cold water; 10Ph, similar to 10P but with pomace soaked in boiling water; WS, dough/bread of wholegrain spelt flour. ^2^ Mean values (± standard deviation) in a column with different superscripts differ significantly (*p* < 0.05). ^3^ Mean values (± standard deviation) in this column with different superscripts differ significantly (*p* < 0.10).

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
