# Peer review of "Pre-Hydrated Berry Pomace in Wheat Bread: An Approach Considering Requisite Water in Fiber Enrichment"

_foods, 2020, doi:10.3390/foods9111600_

Round 1
Reviewer 1 Report
The manuscript entitled ‘Pre-hydrated berry pomace in wheat bread: an approach considering its water binding capacity’ is well-written, contains interesting information/results and is based on a sound experimental approach. Overall, this is a good research paper.
consider minor revisions
1/ is the title specific enough? I doubt whether ‘water binding capacity’ should be in the title, I prefer to mention fiber enrichment instead. The article does not go into depth in how water binding takes place and therefore is can mislead other readers.
2/ Line 86: why does wholegrain spelt flour serves as another control for the enrichment of bread with berry pomace? It is not explained why so it now seems that is was only included in the experimental design to have three different dough matrices. Perhaps the terminology of ‘control’ should be reconsidered.
3/ Line 124-125: Can you support this statement with more references? I find it questionable whether dough stickiness is truly independent of fermentation time.
4/ Line 168-169: Please clarify this sentence. It does not match with the previous sentence where is mentioned that an increase of 4.6% water absorption is observed with 10% berry pomace. Do you refer here to another article? If yes, please make this more clear in the sentence itself, and not solely by the citation number.
5/ Table 2: column 6: ‘ Bread volume’ instead of brad volume.
6/ Line 363: Why did you use blackcurrant pomace and not any other berry pomace? Can you define this in the introduction of the paper? Does blackcurrant pomace specifically have beneficial effects or does any other fiber rich fruit by-product could do the deal? In my opinion, the goal of this paper was to define techniques and results on how the assess overall dough and bread quality of fiber-enriched wheat flour bread. Maybe this should also be clear from reading the title?
In the paper, only in the conclusion a critical note has been added, please mention earlier.
Author Response
Reviewer 1
Comments and Suggestions for Authors
The manuscript entitled ‘Pre-hydrated berry pomace in wheat bread: an approach considering its water binding capacity’ is well-written, contains interesting information/results and is based on a sound experimental approach. Overall, this is a good research paper.
Thank you for your suggestions, we took the opportunity to improve the manuscript.
consider minor revisions
1/ is the title specific enough? I doubt whether ‘water binding capacity’ should be in the title, I prefer to mention fiber enrichment instead. The article does not go into depth in how water binding takes place and therefore is can mislead other readers.
We have reconsidered the title and changed it to “Pre-hydrated Berry Pomace in Wheat Bread: An Approach Considering Requisite Water in Fiber Enrichment”
2/ Line 86: why does wholegrain spelt flour serves as another control for the enrichment of bread with berry pomace? It is not explained why so it now seems that is was only included in the experimental design to have three different dough matrices. Perhaps the terminology of ‘control’ should be reconsidered.
While designing the study different flours were scanned with respect to a proximate composition that is similar to 10P. For that reason we decided to include wholegrain spelt flour to have a “less perfect” comparison than wheat flour. The corresponding text section is rephrased (Section 2.2., lines 93-94).
3/ Line 124-125: Can you support this statement with more references? I find it questionable whether dough stickiness is truly independent of fermentation time.
It is also our own experience. In a previous paper (10.1016/j.lwt.2019.03.069), referring to dough texture properties at different proofing times, the stickiness is therefore displayed at only one time. Sure, directly after kneading when hydration is not yet completed, stickiness may vary. Therefore the measurements were performed after a sufficient resting period. Information is now added to line 134.
4/ Line 168-169: Please clarify this sentence. It does not match with the previous sentence where is mentioned that an increase of 4.6% water absorption is observed with 10% berry pomace. Do you refer here to another article? If yes, please make this more clear in the sentence itself, and not solely by the citation number.
Sentence deleted as recommended by reviewer 3.
5/ Table 2: column 6: ‘ Bread volume’ instead of brad volume.
Typo corrected
6/ Line 363: Why did you use blackcurrant pomace and not any other berry pomace? Can you define this in the introduction of the paper? Does blackcurrant pomace specifically have beneficial effects or does any other fiber rich fruit by-product could do the deal? In my opinion, the goal of this paper was to define techniques and results on how the assess overall dough and bread quality of fiber-enriched wheat flour bread. Maybe this should also be clear from reading the title?
Sentence added to end of introduction and conclusion
In the paper, only in the conclusion a critical note has been added, please mention earlier.
Section added (lines 204-207)
Reviewer 2 Report
Manuscript FOODS-967366 addresses a very topical topic, as is the case of reducing the residues generated by the food industry by using them in the formulation of foods with improved nutritional properties. In this way, the results obtained in this research contribute to meeting the EU Sustainable Development Goals 3 and 12 target to “Ensure healthy lives and promote well-being for all at all ages” and to “Ensure sustainable consumption and production patterns”, respectively. Furthermore, unlike other investigations aimed at recuperating specific components with potentially high market value or obtaining biofuels, what is intended in this work is the comprehensive use of all the waste generated during the blackcurrant juice production.
It is evident from the information found in different databases, that the research group sending this article proposal has experience in formulating doughs with fiber-rich ingredients and in understanding how their water and oil binding capacities can be affected by its interactions with other ingredients and/or the different pre-treatments applied. This is evidenced by an appropriate experimental design, the correct selection of analytical measurements and conditions and the excellent discussion of the results obtained. There are very few comments that could be made to improve the quality of the document under review.
Page 2, line 73: consider explaining the meaning of variable x50,3 since its relationship with the average size of the powder’s particles may not be so obvious.
Page 2, line 78: consider separating the units from the number with a space.
Page 2, table 1: consider including standard deviation values, as well as the degree of significance of the differences observed between columns for each of the components analysed.
Page 6, lines 219 and 220: check that the reference is reported properly, according to the journal’s rules.
Tables 2 and 3: Why are some values reported with 2 decimal figures and others with only one? This would be possible if the criterion of significant figures were applied, but it seems that this is not the case. Please modify tables 2 and 3 according to this criterion.
Author Response
Reviewer 2
Comments and Suggestions for Authors
Manuscript FOODS-967366 addresses a very topical topic, as is the case of reducing the residues generated by the food industry by using them in the formulation of foods with improved nutritional properties. In this way, the results obtained in this research contribute to meeting the EU Sustainable Development Goals 3 and 12 target to “Ensure healthy lives and promote well-being for all at all ages” and to “Ensure sustainable consumption and production patterns”, respectively. Furthermore, unlike other investigations aimed at recuperating specific components with potentially high market value or obtaining biofuels, what is intended in this work is the comprehensive use of all the waste generated during the blackcurrant juice production.
It is evident from the information found in different databases, that the research group sending this article proposal has experience in formulating doughs with fiber-rich ingredients and in understanding how their water and oil binding capacities can be affected by its interactions with other ingredients and/or the different pre-treatments applied. This is evidenced by an appropriate experimental design, the correct selection of analytical measurements and conditions and the excellent discussion of the results obtained. There are very few comments that could be made to improve the quality of the document under review.
Thank you for the kind assessment and the suggestions. We took the opportunity to improve the manuscript.
Page 2, line 73: consider explaining the meaning of variable x50,3 since its relationship with the average size of the powder’s particles may not be so obvious.
It has been specified in line 80.
Page 2, line 78: consider separating the units from the number with a space.
Typo corrected
Page 2, table 1: consider including standard deviation values, as well as the degree of significance of the differences observed between columns for each of the components analysed.
The values for the flours were given by the suppliers and deviations were not provided. Proximate composition of blackcurrant pomace with deviations is published in DOI 10.1002/jsfa.9302. 90% flour + 10% pomace are calculated values. To keep the table consistent, Table 1 remains unchanged.
Page 6, lines 219 and 220: check that the reference is reported properly, according to the journal’s rules.
Done
Tables 2 and 3: Why are some values reported with 2 decimal figures and others with only one? This would be possible if the criterion of significant figures were applied, but it seems that this is not the case. Please modify tables 2 and 3 according to this criterion.
Values of bread volume and crumb moisture were extended to 2 decimals (Table 2). Both Tables contain superscripts indicating significances.
Reviewer 3 Report
Major comments:
The authors studied the rheological and chemical properties of bread dough incorporated with blackcurrant pomace and the quality of their baked products. They have included three pomace preparation methods with and without pre-hydration at different temperatures for their investigation. The purpose of the study was defined properly in the introduction. However, the authors enumerated the results and the discussion of each analysis but did not discuss the study as a whole. Therefore, I recommended the authors to separate the results and the discussion. In this way, the authors are able to discuss for example the rheological properties of dough with pomace (10P, 10Pc and 10Ph) according to the different methods (farinograph, small amplitude oscillation test and extensograph) instead of discussing each method, and compare with Ref. Moreover, the authors should consider omitting the results of WS dough/bread. The reasons are 1) gluten quality of spelt is usually different from bread wheat, therefore it is difficult to separate the effect of the flour composition or the gluten quality 2) not all measurements were done for WS dough (no results for small amplitude oscillation test and changes in dough volume during proofing), 3) even the composition of wholemeal flour and 10P is similar (Table 1), rheological properties of the dough (Fig 3) and the quality of the bread are quite different as the authors also mentioned in their manuscript, 4) there are not much comparison done between 10P and WS. In the discussion or conclusion, the authors should summarize the optimal method for incorporating pomace in wheat dough that gives the best bread quality. At last, the abbreviation was not used consistently through the manuscript, and it should be improved.
Minor comments:
Abstract:
Line 10-14 The authors should rephrase “by additionally adjusting the water level on dough consistency”, as it is true for 10P, but not for 10Pc and 10Ph. The authors adjusted water absorption for 10P, but not for 10Pc and 10Ph. According to the study, the authors compared 1) the effect of incorporation of pomace (Ref vs 10P) and 2) the effect of pre-processing (with and without pre-hydration with different temperatures) of pomace (10P vs 10Pc and 10Ph) on the dough rheology and the quality of baked products.
Line 15-18 Instead of enumerate the results as “ affect pH, proofing and texture” and “differences in complex modulus and loss tangent turned out to be less pronounced”, describe how it affects and what the results means (interpretation of the results).
Line 18-20 The results presented here is very specific and not easy for readers to understand without reading the whole manuscript. I suggest moving to the conclusion.
Line 21-22 The sentence is vague. Moreover, I do not think it is worth mentioning in the abstract, but I recommend discussing in the R&D.
Introduction:
Paragraph 1 (line 29-35) and 2 (line 336-345) should be combined as the authors mentioned the technological challenges in lines 34-35, and detailed technological challenges were explained in the following paragraph.
Line 38 Remove increase from “a reduced bread increase…”
Line 42 Change “raised bread volume” to increased bread volume and “below even 1%” to even below 1%.
Line 40-43 Could you include the amount of soluble fiber (i.e. inulin, pectin and β-glucan)?
Line 54 What do you mean by “only short time scales after water addition were considered in analysis”? Rephrase it and make it clear.
Line 55 What kind of “seed” are you talking about?
Line 56-57 Rephrase to “When hot water was used, enzyme activities were reduced in pre-hydrated grains”
Line 58-59 Could you explain briefly “the model dough system” that is used in your previous study [5]?
Line 59-61 I assume that this is results from your previous study [5]. If so, the reference should be included.
Material and methods
Line 68 I assume that wheat four type 550 is for bread making, but authors should mention it for those who are not familiar with the wheat classification system in Germany.
Line 64/71 Why did you use wholegrain spelt instead of wholemeal of same wheat type? How is the gluten quality in spelt and wheat flour type 550 used in the study? The difference in gluten quality could influence both the rheological properties and the bread quality.
Line 73 What is X50,3?
Line 89 Why do you choose 600 BU instead of 500 BU which is the standard for ICC 115/1?
Line 91 “using the same amount of water as it was determined by water absorption for 10P.” According to the results, waster absorption for Ref was 56.5 %, while that for 10P was 59.1 %. How much water was used for pre-hydration and how much was added when you run Farinograph? I assume that the sum of water used for pre-hydration and Farinograph was 59.1g/100g flour for 10Pc and 10Ph, but it is not clear.
Line 94-96 “first blending the flour or flour/pomace mixture for 1 min at 63 rpm.” The method sounds like it is for pomace without pre-hydration. Or did you also mix the wheat flour and pre-hydrated pomace for 1 min at 63 rpm? And how did you decide the time (4 min) for kneading? Is it development time for Ref?
Line 105-106 “Changes in volume” instead of “Volume development”
Line 124 Why is the duration of main proofing 75 min, not 65 min?
Line 127 How did you prepare the dough without yeast? With farinograph as described in 2.2?
Line 126-134 You can describe what (which parameters) you have measured with instrument and briefly describe what complex modulus (G*) and the loss tangent are and how they are calculated in the M&M. How many measurements did you run, twice?
Line 132 Move “25 °C” later in the sentence (before “,”).
Line 136 The authors prepared four loaves in 2.2 and two of them were used for dough analyses. Please define how you prepared the eight loaves. How do you calculate relative baking loss, difference in weight before and after baking?
Line 158-159 Could you briefly describe what are the hue angle and chroma indicate?
Line 160-163 Authors had different numbers of biological replicates and technical replicates. Please explain how you did handle these biological and technical replicates.
Results and discussion
Line 168-172 Do you really need to mention this? If yes, the first sentence is a bit confusing even though the authors refer [5]. I recommend adding “in the model dough system in our previous study” at the end of the sentence. And the second sentence should include reference nr [5].
Line 173-174 Is the author explaining why wholemeal flour has a high-water affinity? How about fiber?
Line 177-178 It is a bit confusing. Did you knead dough for 4 min or until dough development time?
Line 181-182 I agree with “This means that pre-hydration could be beneficial for dough formation.” But, the results also indicated that 10Pc and 10Ph has even higher water absorption as dough resistance was much higher than 600 BU. Why did not the authors adjust the water level for 10Pc and 10Ph to obtain dough resistance at 600 BU?
Line 184 Which water absorption was used for WS dough?
Line 205-206 Please use abbreviation consistently. You have already defined 10Pc and 10Ph in the M&M.
Line 219 Combine with the previous paragraph (Line 214-218).
Line 220 Check the citation style for “Schmidt et al., 2018”
Line 224 Remove “Ref” as you are discussing the dough with pomace (?)
Line 230 Change to Increase in dough volume instead of “dough volume development”. Same for the capture in fig 2.
Fig 2. I do not understand the equation especially “ml/min3”, “ml/min2” and “ml/min”. Are they part of the coefficient?
Line 238-239 “The second power exponent”, are you talking about the coefficient (0.033 and 0.024 ml/min2) for min2 here? Could you make sure if you are using the right term?
Line 242-244 I do not understand the sentence. Please rephrase it.
Line 244 Rephrase “reduce volume growth”.
Line 248-249 What do you mean by “during the main rise up”?
Line 252 Remove second “presumably” from the sentence.
Line 253-254 Could you explain what kind of “interaction” you are discussing here? Remove “see” before Figure 1 (same for line 271, 299, 328…)
Line 257 What do you mean by “gluten formation”? How can reason (a) explain the changes in dough resistance from high after pre-proofing to low after main proofing? If reason (b) is true, how can you explain the highest dough resistance after pre-proofing in 10Ph?
Line 271-272 Usually, low resistance and high extensibility = weaker gluten and high dough resistance = strong gluten.
Line 273 I do not understand “a restricted molecular mobility through the formation of fiber/protein interactions”
Line 280 Instead of “an enhanced wetting contact between dough surface and the measuring plug”, couldn’t you simply describe that Ref and WS doughs were stickier than 10P dough?
Line 293 What is the “chemical process”?
Line 295 increased instead of “intensified”
Line 300 Not “beyond” but during.
Line 304 Instead of “Water binding through baking was lower for WS, “, could you simply write the baking loss was pronounced for WS or WS lost more water during baking, (or something?)
Line 308 The authors wrote that gluten is less resistance in WS, however line 271-272 they wrote WS has strong gluten network after Ref. Please check the comments for the line 271-272 and be consistent.
Line 309-10 Please use abbreviations consistently. Not “enhanced” but “increased” bread volume. And make sure what you compare the bread volume with (10P or WS?). What do you mean by “produce negligible effect”?
Line 311 Is it “pomace and isolated fiber” or pomace or isolated fiber?
Line 313 Change “as shown” to as shown in this study to make this clear
Line 314-315 Please check abbreviation
Line 316 I agree that average crumb firmness is much higher for WS then other bread with/without pomace, however, is was not significant because of large S.D. The authors should discuss or at least mention why S.D. is so big for WS bread (for both crust and crumb firmness).
Line 334 It is true. However, the relationship between crumb firmness and the total cell area is not as straight forward when the crumb firmness and the total cell area were compared between 10P, 10Pc and 10Ph.
Line 335 Could you briefly describe how pH (high or low) affect gas production rate (high or low), bread volume and crumb firmness (positive or negative)? Since your dough pH decrease with pomace, you could mention how low pH affect these parameters according to ref [36].
Line 336-337 Is it your results?
Line 337-340 What is the connection with the previous sentence? This part reads conclusion. I recommended to move to the conclusion and describe a bit more than “hydration properties are considered”. Do the authors recommend pre-hydration of pomace? If yes, which temperature? How about water absorption?
Line 346-351 and Figure 4 The bread with pomace (10P) showed dark garish colour. Even though authors did not study the consumer acceptance of the bread, you could discuss if/how the bread would be accepted by consumers by referring others study without speculating too much.
Line 358-362 This part is supposed to be the conclusion but reads like results and discussion. See the comment for line 337-340.
Line 363-364 Is it really the main outcome of the study? How is it relevant for research? Could you give some specific example?
Line 364-366 This part does not read like the conclusion, move to the end of discussion.
Author Response
Reviewer 3
Comments and Suggestions for Authors
Major comments:
The authors studied the rheological and chemical properties of bread dough incorporated with blackcurrant pomace and the quality of their baked products. They have included three pomace preparation methods with and without pre-hydration at different temperatures for their investigation. The purpose of the study was defined properly in the introduction. However, the authors enumerated the results and the discussion of each analysis but did not discuss the study as a whole. Therefore, I recommended the authors to separate the results and the discussion. In this way, the authors are able to discuss for example the rheological properties of dough with pomace (10P, 10Pc and 10Ph) according to the different methods (farinograph, small amplitude oscillation test and extensograph) instead of discussing each method, and compare with Ref. Moreover, the authors should consider omitting the results of WS dough/bread. The reasons are 1) gluten quality of spelt is usually different from bread wheat, therefore it is difficult to separate the effect of the flour composition or the gluten quality 2) not all measurements were done for WS dough (no results for small amplitude oscillation test and changes in dough volume during proofing), 3) even the composition of wholemeal flour and 10P is similar (Table 1), rheological properties of the dough (Fig 3) and the quality of the bread are quite different as the authors also mentioned in their manuscript, 4) there are not much comparison done between 10P and WS. In the discussion or conclusion, the authors should summarize the optimal method for incorporating pomace in wheat dough that gives the best bread quality. At last, the abbreviation was not used consistently through the manuscript, and it should be improved.
Thank you for the suggestions. We took the opportunity to improve the manuscript.
We thoroughly discussed about using WS and came to the following conclusion. (1) We decided to choose wheat flour as reference, as part of it was then replaced by pomace. (2) As the use of pomace enriches dietary fiber but dilutes gluten content it can be expected that product properties will vary from Ref. (3) Before executing the study different flours were scanned with respect to a proximate composition similar to the 10P formulation. This was the reason why we decided to include wholegrain spelt flour to have a “less perfect” comparison. (4) The significant differences between pomace formulations and WS are experimental results and could not be expected in advance.
To conclude on this comment, discussion and conclusion of the manuscript are now extended (for details see comments below). All abbreviations are checked with respect to consistency.
Minor comments:
Abstract:
Line 10-14 The authors should rephrase “by additionally adjusting the water level on dough consistency”, as it is true for 10P, but not for 10Pc and 10Ph. The authors adjusted water absorption for 10P, but not for 10Pc and 10Ph. According to the study, the authors compared 1) the effect of incorporation of pomace (Ref vs 10P) and 2) the effect of pre-processing (with and without pre-hydration with different temperatures) of pomace (10P vs 10Pc and 10Ph) on the dough rheology and the quality of baked products.
Thank you for the comment. The water absorption according ICC standard 115/1 was done in preliminary experiments (line 95). Following this method only flour (or flour/pomace mixture) corrected to 14% moisture is kneaded with water. Flour/pomace composition of 10P, 10Pc, 10Ph is similar and therefore the amount of water was chosen identical. The parameters obtained from the resulting farinograms were assigned to bread making procedure. Lines 12-15 are rephrased.
Line 15-18 Instead of enumerate the results as “ affect pH, proofing and texture” and “differences in complex modulus and loss tangent turned out to be less pronounced”, describe how it affects and what the results means (interpretation of the results).
The section was revised as suggested. See lines 16-22.
Line 18-20 The results presented here is very specific and not easy for readers to understand without reading the whole manuscript. I suggest moving to the conclusion.
The section was revised as suggested. See lines 16-22.
Line 21-22 The sentence is vague. Moreover, I do not think it is worth mentioning in the abstract, but I recommend discussing in the R&D.
Precise information was added (line 23)
Introduction:
Paragraph 1 (line 29-35) and 2 (line 336-345) should be combined as the authors mentioned the technological challenges in lines 34-35, and detailed technological challenges were explained in the following paragraph.
The first section is a general introduction and paragraph 2 focuses already on details. The parts remain divided in units of thoughts.
Line 38 Remove increase from “a reduced bread increase…”
Done
Line 42 Change “raised bread volume” to increased bread volume and “below even 1%” to even below 1%.
Done
Line 40-43 Could you include the amount of soluble fiber (i.e. inulin, pectin and β-glucan)?
The amount for pectin and beta-glucan (1%) was added to line 45.
Line 54 What do you mean by “only short time scales after water addition were considered in analysis”? Rephrase it and make it clear.
The sentence was specified in lines 56,57: i.e. during kneading or after a resting period of 10 min
Line 55 What kind of “seed” are you talking about?
Customary seeds such as sunflower seeds, chia, linseed, but also grains and groats.
Line 56-57 Rephrase to “When hot water was used, enzyme activities were reduced in pre-hydrated grains”
Sentence changed as suggested
Line 58-59 Could you explain briefly “the model dough system” that is used in your previous study [5]?
The model dough contained only flour, water, and pomace. (line 63)
Line 59-61 I assume that this is results from your previous study [5]. If so, the reference should be included.
You are right. Reference added
Material and methods
Line 68 I assume that wheat four type 550 is for bread making, but authors should mention it for those who are not familiar with the wheat classification system in Germany.
line 75: “all-pupose” was added
Line 64/71 Why did you use wholegrain spelt instead of wholemeal of same wheat type? How is the gluten quality in spelt and wheat flour type 550 used in the study? The difference in gluten quality could influence both the rheological properties and the bread quality.
As stated above, different flours were scanned with respect to a proximate composition similar to 10P. Different gluten properties are apparent and were not analysed.
Line 73 What is X50,3?
It is the volume based median, describing the particle size. Half of the particles are below and half above the given value. Specified in line 80.
Line 89 Why do you choose 600 BU instead of 500 BU which is the standard for ICC 115/1?
The dough prepared at 500 BU was very sticky and unable to process. We were wondering about it but, after a new calibration of the farinograph, the “problem” remained. The Arbeitsgemeinschaft Getreideforschung e.V. (Detmold, Germany) told us in a conversation that the flours have changed since the method was developed and an adaption is appropriate.
Line 91 “using the same amount of water as it was determined by water absorption for 10P.” According to the results, water absorption for Ref was 56.5 %, while that for 10P was 59.1 %. How much water was used for pre-hydration and how much was added when you run Farinograph? I assume that the sum of water used for pre-hydration and Farinograph was 59.1g/100g flour for 10Pc and 10Ph, but it is not clear.
Right, we added the value in line 99.
Line 94-96 “first blending the flour or flour/pomace mixture for 1 min at 63 rpm.” The method sounds like it is for pomace without pre-hydration. Or did you also mix the wheat flour and pre-hydrated pomace for 1 min at 63 rpm? And how did you decide the time (4 min) for kneading? Is it development time for Ref?
The first blending was only applicable for dry flour and pomace. Pre-hydrated pomace was added with the water after the 1 min. See line 103/104. Kneading time is based on Ref dough development time + half time of dough stability (3 min + 1 min).
Line 105-106 “Changes in volume” instead of “Volume development”
Changed as suggested
Line 124 Why is the duration of main proofing 75 min, not 65 min?
After 65 min dough extensibility measurements were performed. Therefore stickiness was measured chronological.
Line 127 How did you prepare the dough without yeast? With farinograph as described in 2.2?
Exactly (line 136)
Line 126-134 You can describe what (which parameters) you have measured with instrument and briefly describe what complex modulus (G*) and the loss tangent are and how they are calculated in the M&M. How many measurements did you run, twice?
Storage modulus (G’) and loss modulus (G’’) were derived from the measurements. Complex modulus G* describes the overall resistance to deformation and indicates stiffness of the dough. It is calculated from the Pythagoras of G’ and G’’. Loss tangent (tan δ) is the quotient of G’’ and G’ and indicates whether elastic or viscous contributions are predominating. Measurements were performed on two individually prepared doughs (n=2). (lines 145-149)
Line 132 Move “25 °C” later in the sentence (before “,”).
Done
Line 136 The authors prepared four loaves in 2.2 and two of them were used for dough analyses. Please define how you prepared the eight loaves. How do you calculate relative baking loss, difference in weight before and after baking?
The individual dough mass was >480 g. For dough texture analysis 3 x 20 g were taken, and loaves of 4 x 100 g were baked. Two doughs were prepared respectively, resulting in 8 loaves. Results in Figure 2 were derived from separate doughs.
Baking loss (%) = (1- (mass of baked loaf)/(mass of shaped dough piece))*100
Line 158-159 Could you briefly describe what are the hue angle and chroma indicate?
Added to line 174
Line 160-163 Authors had different numbers of biological replicates and technical replicates. Please explain how you did handle these biological and technical replicates.
All data (biological and technical replicates) were pooled before calculating mean +/- standard deviations. These data were also used for ANOVA calculations. For example, n=8 for bread volume in Table 2: Two independent dough preparations and baking, and four bread loaves from each dough gives a total of n=8 replicates.
Results and discussion
Line 168-172 Do you really need to mention this? If yes, the first sentence is a bit confusing even though the authors refer [5]. I recommend adding “in the model dough system in our previous study” at the end of the sentence. And the second sentence should include reference nr [5].
Section deleted
Line 173-174 Is the author explaining why wholemeal flour has a high-water affinity? How about fiber?
Frakolaki et al. 2018 accounted the high water absorption and stickiness of spelt flour to overall high protein content and a higher gliadin/glutenin ratio opposed to wheat flour. They did not discuss the impact of dietary fiber in detail. Sentence added (line 188)
Line 177-178 It is a bit confusing. Did you knead dough for 4 min or until dough development time?
Clause added to be more clear (line 192)
Line 181-182 I agree with “This means that pre-hydration could be beneficial for dough formation.” But, the results also indicated that 10Pc and 10Ph has even higher water absorption as dough resistance was much higher than 600 BU. Why did not the authors adjust the water level for 10Pc and 10Ph to obtain dough resistance at 600 BU?
The proposal is interesting, but we think this would be a totally different approach. Our intention was using the same water content in pomace formulations, but different hydration conditions.
Sentences added (lines 204-207)
Line 184 Which water absorption was used for WS dough?
ICC 115/1, consistent for all flours & flour/pomace mixture
Line 205-206 Please use abbreviation consistently. You have already defined 10Pc and 10Ph in the M&M.
Done
Line 219 Combine with the previous paragraph (Line 214-218).
Done
Line 220 Check the citation style for “Schmidt et al., 2018”
Done
Line 224 Remove “Ref” as you are discussing the dough with pomace (?)
Sentence changed to: The corresponding temperature at maximum G* tended to decrease in the order Ref, 10P, 10Pc, 10Ph (87.3 – 84.0 °C), pointing on reduced water competition with starch granules when hydrated pomace was incorporated (lines 241-243).
Line 230 Change to Increase in dough volume instead of “dough volume development”. Same for the capture in fig 2.
Changed as suggested.
Fig 2. I do not understand the equation especially “ml/min3”, “ml/min2” and “ml/min”. Are they part of the coefficient?
The cubic equation (y = a x³ + b x² + c x +d) has the coefficients a, b, c, d
y is the dough volume (mL) and x refers to the proofing time (min). Instead using “x” in the equation the individual units were used. For example, in the quadratic term, x² comes with the unit time², in our case min². For that purpose, the unit of the coefficient must be mL/min² because dough volume as such (left side of equation) also is in volume units.
Line 238-239 “The second power exponent”, are you talking about the coefficient (0.033 and 0.024 ml/min2) for min2 here? Could you make sure if you are using the right term?
Thank you, we replaced by „quadratic term“.
Line 242-244 I do not understand the sentence. Please rephrase it.
Subclause added to be more clear (line 261)
Line 244 Rephrase “reduce volume growth”.
Changed to “impaired increase in volume” (line 263)
Line 248-249 What do you mean by “during the main rise up”?
Comma inserted to be more clear “During the main rise, up to 65 min total proofing time,”
Line 252 Remove second “presumably” from the sentence.
Done
Line 253-254 Could you explain what kind of “interaction” you are discussing here? Remove “see” before Figure 1 (same for line 271, 299, 328…)
Resistance is favoured by hydrogen bonds between pomace and the gluten, but also physical interactions through fiber particles are accountable [5]
“see” was deleted in all cases
Line 257 What do you mean by “gluten formation”? How can reason (a) explain the changes in dough resistance from high after pre-proofing to low after main proofing? If reason (b) is true, how can you explain the highest dough resistance after pre-proofing in 10Ph?
The explanation was in the wrong order (now changed). Denatured proteins increased resistance opposed to 10P and 10Pc. The available water in the matrix had positive impact on gluten structure and with proceeding proofing relaxation processes were favoured.
Line 271-272 Usually, low resistance and high extensibility = weaker gluten and high dough resistance = strong gluten.
Indeed, but not in this context. An interrupted structure can tear easily and fiber may increase resistance. Most authors measure extensibility of dough without yeast and at one time.
Line 273 I do not understand “a restricted molecular mobility through the formation of fiber/protein interactions”
See comment to “Line 253-254” above. Clause added to line 293
Line 280 Instead of “an enhanced wetting contact between dough surface and the measuring plug”, couldn’t you simply describe that Ref and WS doughs were stickier than 10P dough?
Done as suggested
Line 293 What is the “chemical process”?
Protein denaturation, starch gelatinization, yeast & enzyme inactivation, …
Line 295 increased instead of “intensified”
Changed
Line 300 Not “beyond” but during.
Changed
Line 304 Instead of “Water binding through baking was lower for WS, “, could you simply write the baking loss was pronounced for WS or WS lost more water during baking, (or something?)
Sentence rephrased. “WS lost more water during baking, since crumb moisture was identical to 10Ph despite the highest water absorption of wholegrain spelt flour during kneading.” line 325
Line 308 The authors wrote that gluten is less resistance in WS, however line 271-272 they wrote WS has strong gluten network after Ref. Please check the comments for the line 271-272 and be consistent.
Clause rephrased “baking properties of its gluten composition are different”
Still thinking it is more precise to write gluten is less resistant.
Line 309-10 Please use abbreviations consistently. Not “enhanced” but “increased” bread volume. And make sure what you compare the bread volume with (10P or WS?). What do you mean by “produce negligible effect”?
Revised to be more clear (lines 328-329)
Line 311 Is it “pomace and isolated fiber” or pomace or isolated fiber?
“or” instead of “and”; changed
Line 313 Change “as shown” to as shown in this study to make this clear
Done
Line 314-315 Please check abbreviation
Done
Line 316 I agree that average crumb firmness is much higher for WS then other bread with/without pomace, however, is was not significant because of large S.D. The authors should discuss or at least mention why S.D. is so big for WS bread (for both crust and crumb firmness).
Please excuse, it was a typo. Correct value 13.71 ± 3.32
Line 334 It is true. However, the relationship between crumb firmness and the total cell area is not as straight forward when the crumb firmness and the total cell area were compared between 10P, 10Pc and 10Ph.
I agree. The air in the cells is the same, but the connections between the cells differ in firmness and cohesion.
Line 335 Could you briefly describe how pH (high or low) affect gas production rate (high or low), bread volume and crumb firmness (positive or negative)? Since your dough pH decrease with pomace, you could mention how low pH affect these parameters according to ref [36].
Section added (lines 356-359): “In detail, lower pH tended to slow down yeast activity and smaller cells were formed. This resulted in a higher cell density. As the acidity of dough was still appropriate, high number of cells was formed during fermentation.”
Line 336-337 Is it your results?
Yes
Line 337-340 What is the connection with the previous sentence? This part reads conclusion. I recommended to move to the conclusion and describe a bit more than “hydration properties are considered”. Do the authors recommend pre-hydration of pomace? If yes, which temperature? How about water absorption?
Mentioned section moved to conclusions. Conclusion was extended.
Line 346-351 and Figure 4 The bread with pomace (10P) showed dark garish colour. Even though authors did not study the consumer acceptance of the bread, you could discuss if/how the bread would be accepted by consumers by referring others study without speculating too much.
Section added (lines 372-377)
Line 358-362 This part is supposed to be the conclusion but reads like results and discussion. See the comment for line 337-340.
This section summarizes the study. So it fits less in discussion.
Line 363-364 Is it really the main outcome of the study? How is it relevant for research? Could you give some specific example?
Conclusion was extended.
Line 364-366 This part does not read like the conclusion, move to the end of discussion.
Think we have different views what should be in a conclusion.
Round 2
Reviewer 3 Report
The manuscript has been improved, however there are some points that the authors should consider/amend:
Line 89 Why do you choose 600 BU instead of 500 BU which is the standard for ICC 115/1?
>The dough prepared at 500 BU was very sticky and unable to process. We were wondering about it but, after a new calibration of the farinograph, the “problem” remained. The Arbeitsgemeinschaft Getreideforschung e.V. (Detmold, Germany) told us in a conversation that the flours have changed since the method was developed and an adaption is appropriate.
I suggest including “with slight modification” after “ICC standard 115/1 [16]” in line 141.
Line 160-163 Authors had different numbers of biological replicates and technical replicates. Please explain how you did handle these biological and technical replicates.
>All data (biological and technical replicates) were pooled before calculating mean +/- standard deviations. These data were also used for ANOVA calculations. For example, n=8 for bread volume in Table 2: Two independent dough preparations and baking, and four bread loaves from each dough gives a total of n=8 replicates.
Does it mean that the authors treat the biological and technical replicates at the same level? An adequate measure of uncertainty should be based on the biological replicates and not from the technical replicates that shows the variation caused by measurement. Have the authors considered this?
Fig 2. I do not understand the equation especially “ml/min3”, “ml/min2” and “ml/min”. Are they part of the coefficient?
>The cubic equation (y = a x³ + b x² + c x +d) has the coefficients a, b, c, d
y is the dough volume (mL) and x refers to the proofing time (min). Instead using “x” in the equation the individual units were used. For example, in the quadratic term, x² comes with the unit time², in our case min². For that purpose, the unit of the coefficient must be mL/min² because dough volume as such (left side of equation) also is in volume units. The cubic equation (y = a x³ + b x² + c x +d) has the coefficients a, b, c, d
The formula showed in the figure is not the cubic equation. It is because for example the formula for (Ref), y= -0.0004 ml/min3* min3 + 0.033 ml/min2* min2 + 0.07 ml/min*min + 30.03 = -0.0004 ml+ 0.033 ml+ 0.07 ml+ 30.03. I do not know if the coefficient usually has a unit, but the unit presented in the formula caused the confusion. Therefore, I suggest omitting “ml/min3”, “ml/min2” and “ml/min” from the formula, i.e. (Ref), y= -0.0004 *min3 + 0.033 *min2 + 0.07 *min + 30.03.
Line 316 I agree that average crumb firmness is much higher for WS then other bread with/without pomace, however, is was not significant because of large S.D. The authors should discuss or at least mention why S.D. is so big for WS bread (for both crust and crumb firmness).
>Please excuse, it was a typo. Correct value 13.71 ± 3.32
Now I am a bit sceptical … if it was a typo, why did not the ANOVA result show a significant difference between WS and Ref, 10P? Additionally, the S.D. for crust firmness of WS (± 13.30) is much higher than for others (Ref, 10P, 10Pc and 10Ph). Is it also a typo? I suggest that the authors should check the raw data and carry out the statistical analysis again.
